# Functional dynamics between resident transcriptionally active microbes (TAMs) and host genes underlie Dengue severity

Pallawi Kumari[1,2], Priti Devi[1], Basudha Banerjee[1], Bansidhar Tarai[3], Sandeep Budhiraja[3], Tav Pritesh Sethi[2]*, Rajesh Pandey[1,4]*

1 Division of Infectious Disease Biology, INtegrative GENomics of Host-PathogEn (INGEN-HOPE) laboratory, CSIR-Institute of Genomics and Integrative Biology (CSIR-IGIB), Mall Road, Delhi, India, 2 Indraprastha Institute of Information Technology (IIIT) Delhi, New Delhi, India, 3 Max Super Speciality Hospital (A Unit of Devki Devi Foundation), Max Healthcare, Delhi, India, 4 Academy of Scientific and Innovative Research (AcSIR), Ghaziabad, India

* rajeshp.igib@csir.res.in

## Abstract

Host-microbe interactions are increasingly recognized as an important module to understand disease progression and potential treatment strategies. Increasing evidence points to the microbiome's ability to modulate host gene expression, and thereby influencing host physiology. By integrating dual RNA sequencing with machine learning, we uncover how transcriptionally active microbes (TAMs) may influence host genes involved in immune and metabolic functions in the hospital admitted dengue patients. Towards this, we analyzed 112 whole transcriptomes from the blood samples of patients with differential dengue disease severity. Using a machine learning-based integrated host-microbial transcriptomic analysis framework, combining Lasso regression and sparse canonical correlation analysis (SCCA), we identified both shared and disease-specific associations between the microbes and the host transcriptomic pathways. Notably, opportunistic microbes such as *Acetobacter-ghanensis*, *Achromobacter sp. B7*, *Bacillus licheniformis*, and *Clostridium cochlearium*, along with the host genes, namely, *PPME1*, *TIMP2*, *NLRC4*, and *RhoB*, were associated with immune dysregulation in the severe dengue patients. These microbes and genes appear to influence pathophysiology through distinct molecular pathways, highlighting their disease-specific roles in host-microbe interactions.

## Author summary

Dengue represents a wide clinical spectrum, with millions of global cases and thousands of deaths annually. Along with the variable clinical outcomes mild to severe, dengue not only causes subtle symptoms but organ failure also further underscoring the urgent need to uncover driving mechanisms of disease

**Data availability statement:** RNA-Seq data have been deposited and are publicly available in the NCBI Sequence Read Archive (SRA) database under BioProject ID: PRJNA1071729. All relevant data are within the manuscript and its Supporting Information files.

**Funding:** This study received financial support from Bill and Melinda Gates Foundation (BMGF), (Grant number - INV-033578), Rockefeller Foundation, (grant number-2021 HTH 018) and P-CARRS (Grant number – GAP0285) awarded to RP. PK and PD received their salary from the BMGF project. BB received her salary from P-CARRS project. The funders had no role in study design, data collection and analysis, decision to publish, or preparation of the manuscript.

**Competing interests:** The authors have declared that no competing interests exist.

severity. In this study, we have integrated host genes and transcriptionally active microbes with machine learning to analyze 112 blood transcriptomes from dengue patients of varying severity, i.e., mild, moderate and severe. Using LASSO regression and sparse canonical correlation analysis (SCCA), we uncovered both shared and severity-specific host-microbe associations. Opportunistic microbes such as *Acetobacter ghanensis*, *Achromobacter sp. B7*, and *Clostridium cochlearium* were linked to severe dengue. These microbes correlated with immune-related host genes like *PPME1*, *TIMP2*, *NLRC4*, and *RhoB*, indicating immune dysregulation. The study highlights distinct molecular interactions that may underlie severe dengue pathophysiology.

## Introduction

Dengue virus infection is one of the most prevalent mosquito borne infections, endemic to Africa, America, Eastern Mediterranean, South East Asia and Western Pacific [1]. India itself has witnessed a surge in dengue cases since 2019. In 2024 alone, India has reported 1,86,567 incidents and 160 mortality cases as of 31 October 2024 as per National Center for Vector Borne Diseases Control, India [2, 3]. Dengue patients manifest a wide clinical spectrum, ranging from asymptomatic cases to severe manifestations involving multiple organ failure. According to WHO statistics, more than 7.6 million dengue cases were reported till 30th April 2024 including 16,000 severe cases and more than 3000 mortality cases. Delhi has been a hotspot for dengue cases with 52,121 cases and 54 deaths in the last 5 years [4, 5]. The WHO 2009 classification categorizes dengue into three groups: dengue without warning signs (DwoWS), dengue with warning signs (DwWS), and severe dengue (SD) [6]. Severe dengue is marked by complications such as severe bleeding, plasma leakage, and organ impairment ( [7, 8]. This broad range of clinical presentations underscores the need to investigate the underlying differences in pathophysiology that contribute to varying disease severities.

Research into infectious diseases has extensively examined viral and host factors to explain diverse clinical outcomes, yet the underlying immune-pathogenic mechanisms remain unclear and are likely multifactorial [9, 10]. Notably, most studies rely on delineating either the microbiome or host genes separately, which makes host gene-microbiome associations a less intervened area [11–14]. However, it is worth noting that host genes have been reported to play a crucial role in shaping the microbiome by modulating immune responses [14]. Insights from human genetic and genome-wide association studies reveal strong associations between genomic regions, loci, and microbial abundance. Notably, host gene variants can influence microbiome composition directly or indirectly through downstream gene expression, ultimately shaping phenotypes [14]. However, these studies often rely on genome-based microbial detection using conventional approaches. In contrast, our emphasis lies on investigating transcriptionally active microbes (TAMs) through RNA-Seq, which may provide novel insights into their role in modulating disease severity and

outcomes. For example, our studies have shown compositional differences in TAMs associated with the SARS-CoV-2 Pre-VOC, Delta, and Omicron variants, wherein enrichment of *Pseudomonas* and *Acinetobacter* was present in Delta rather than Omicron [14]. Microbe-host gene correlation network also revealed *Acinetobacter baumannii*, *Pseudomonas stutzeri*, and *Pseudomonas aeuroginosa* to be modulating immune pathways, which might augment clinical severity in Delta [15 - 17]. Similarly, in cases of dengue, we discovered that the patients who had high dengue viral load had an elevated level of opportunistic microbial species compared to commensals. Contrarily, the balanced presence of pathogenic/opportunistic and commensals plausibly explains the milder clinical manifestations in the dengue patients with low viral load [9, 18]. This highlights the need for an in-depth investigation of host genes and TAMs to better comprehend the clinical dynamics of patients. The current study utilizes a dual approach via integration of transcriptionally active microbiome data with the host gene expression data to discover hitherto unknown biological associations. A machine learning (ML) framework that comprehensively integrates TAMs, host genes, and biological pathways from 112 hospital admitted dengue patients leverages our understanding of underlying pathophysiological situations.

## Results

### Integrated clinical data and study methodology

We analyzed 112 NS1 antigen positive patients admitted between August-November 2022 at the Max Hospital, Delhi, India. Clinical assessment revealed heterogeneity in the hematological markers commonly associated with dengue differential severity. While the WHO classifies dengue into three categories-without warning signs, with warning signs, and severe, many patients in our cohort did not fully match these criteria. However, for better distinction, we employed a previously published modified classification scheme from the lab: patients with normal platelet and leukocyte counts as mild (n = 45), those with leukopenia but normal platelets as moderate (n = 46), and those with both thrombocytopenia and leukopenia as severe (n = 21) (Fig 1A). This tailored stratification allowed for more precise analytical comparisons and clear disease manifestations in our Indian study population which has high incidences of dengue. The average age of patients taken for this study was 28.5 years wherein male and female patients accounted for 68.8% and 31.2%, respectively. A significant decrease is observed throughout the patient groups for the TLC, platelet counts, and neutrophils. The reduced leukocyte and platelet counts suggest that the virus suppresses the host's immune system, impairing blood cell production while also destroying existing blood cells. Conversely, the increase in lymphocyte levels in moderate and severe groups, compared to the mild cases, aligns with the heightened infection levels and reflects the immune responses generated by the host. According to the liver function tests, total protein and albumin levels were also reduced in proportion to the increased severity. Moreover, SGOT-AST (Serum Glutamic-oxaloacetic transaminase-Aspartate transaminase), a marker of cellular injury in organs such as the liver, heart, and kidneys, tends to show elevated levels in patients with increasing disease severity. All the details of demographic information and clinical parameters for the three patient groups are included in (S1 Table), and key vital parameters have been depicted in Table 1 for emphasis.

RNA sequencing of the 112 patients yielded 18,784,781 raw reads, of which 17,159,379 passed stringent quality control and were mapped to the human genome (S1 File). The remaining 1,625,402 unmapped reads, representing the human-unmapped reads fraction, were retained for potential transcribed microbes profiling. The total raw reads generated from in-house sequencing were aligned to the human genome using HISAT2, yielding mapping rates of 92.92%, 92.62% and 91.78% for the mild, moderate, and severe disease groups, respectively. Upon classification of reads, the distribution of human-mapped versus human-unmapped reads across the severity groups was consistent: mild (91.4% human-mapped, 8.6% human-unmapped reads), moderate (91.3% human-mapped, 8.7% human-unmapped reads), and severe (91.5% human-mapped, 8.5% human-unmapped reads). Separated unmapped (human-unmapped reads) reads were aligned to a comprehensive microbial reference database including bacterial, archaeal, and viral genomes using Kraken2 and Bracken2. Microbial content distribution was relatively uniform across severity groups, comprising 14.7%, 17.7%, and

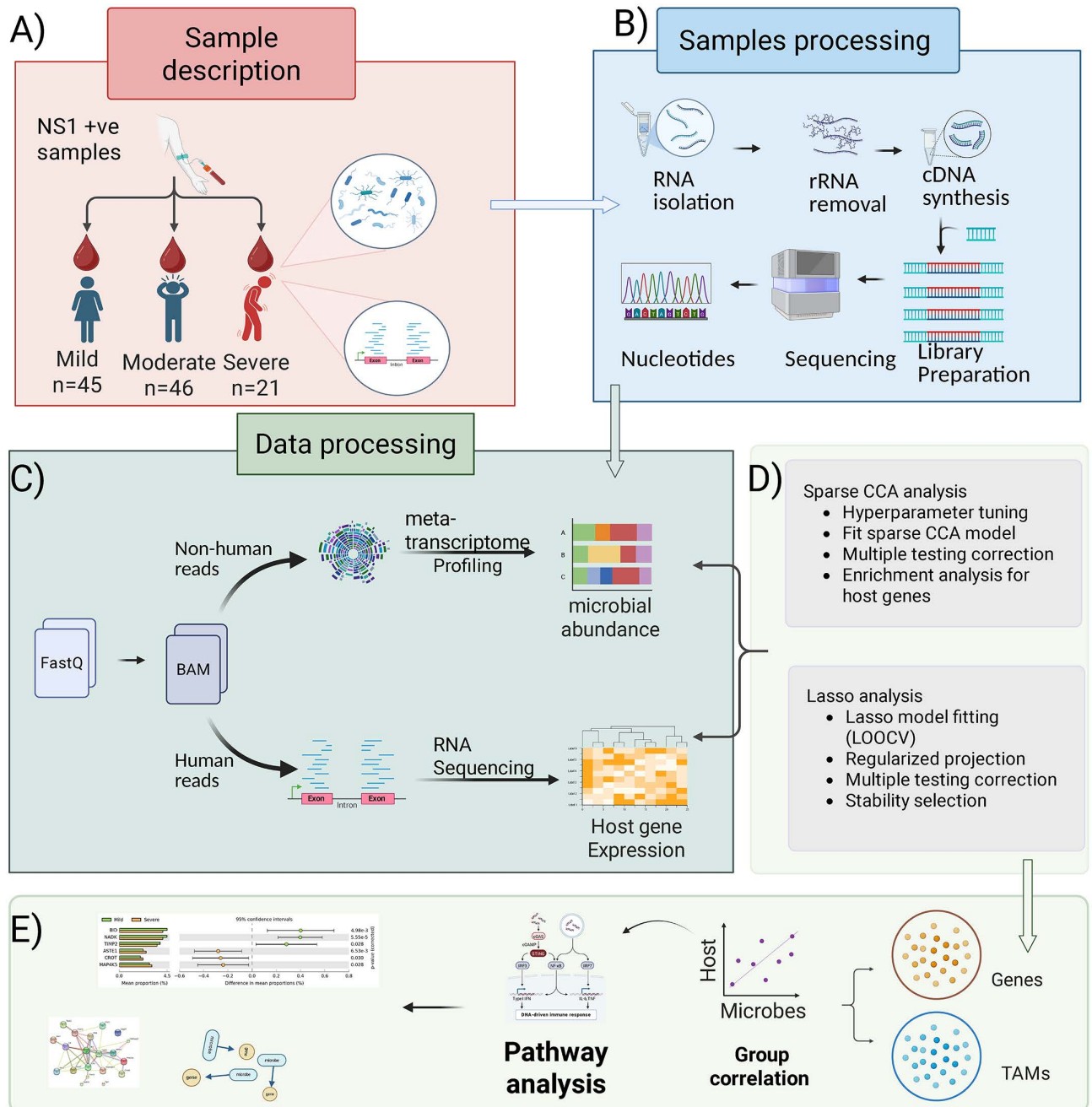

**Fig 1. Overall study design of the study. A)** Samples were collected from patients across different severity groups (mild, moderate, and severe). **B)** Collected samples underwent RNA extraction, quality control, and quantification. **C)** For microbial profiling, meta-transcriptome analysis was performed, while host gene expression was analyzed using RNA sequencing. **D)** SCCA and LASSO analysis for severity-specific host gene-microbe associations. **E)** Enrichment analysis of selected genes indicates immune and inflammatory pathways. Figure created with Biorender.com (Devi, P. (2025) https://BioRender.com/7c9ap3n).

**Table 1. Patient demographics and laboratory markers associated with dengue infection.**

| Dengue Clinical Parameters | Mild (n=45) | Moderate (n=46) | Severe (n=21) | p-value | |
|---|---|---|---|---|---|
| Age | 27 | 26 | 35 | 0.14 | b |
| Gender (F\|M) | 15\|30 | 17\|29 | 4\|17 | 0.34 | a |
| NS1 antigen | 3.22 | 3.5 | 3.5 | 0.75 | b |
| Total Leucocyte Count (TLC) | 5 | 3 | 2.7 | <0.001 | b |
| Platelet Count | 175 | 160 | 125 | <0.001 | b |
| Neutrophil | 75.9 | 63.3 | 62.6 | <0.001 | b |
| Lymphocyte | 14.1 | 24.5 | 26 | <0.001 | b |
| Total Protein | 7.3 | 6.7 | 6.7 | 0.03 | b |
| Albumin | 4.4 | 4 | 4 | 0.05 | b |
| Bilirubin (Total) | 0.7 | 0.4 | 0.75 | 0.02 | b |
| Bilirubin (Direct) | 0.135 | 0.09 | 0.17 | 0.02 | b |
| Bilirubin (Indirect) | 0.56 | 0.31 | 0.55 | 0.03 | b |
| SGOT-Aspartate Transaminase (AST) | 41 | 59 | 97.5 | 0.03 | b |

*a-chi square; b-kruskal wallis*

17.3% of total microbial reads in mild, moderate, and severe cases, respectively (S1 Fig). Specifically, the dengue viral read counts varied across severity groups (Mild: 22,005; Moderate: 4,473; Severe: 5,006). As we have taken samples on the first day when symptomatic patients reported to the hospital for testing, we are unable to pinpoint the exact day of infection. Thus different viral counts between mild and moderate/severe may be reflective of the different time from infection to their reporting to the hospital for testing. At the same time, clinical parameters affirm the differential disease severity.

This dual RNA-Seq approach enables a comprehensive dataset that offers a unique opportunity to analyze host responses alongside microbial community dynamics across different dengue severity. This novel integration includes a machine learning framework for combining high-dimensional information, including microbiome abundance and host gene expression. This approach consists of two components: (1) sparse CCA to identify groups of host genes that associated with groups of TAMs to characterize pathway-level associations, and (2) lasso penalized regression to identify specific associations between the individual host genes and TAMs. To prevent any possible batch effects, the integration analysis was performed independently on the transcribed microbiota and host gene expression data. Prior to using statistical techniques, the host gene count and microbiome abundance data were standardized and normalized in order to meet the statistical models distribution requirements (Fig 1C-E).

## Diversity driven functional profiling of host-TAMs interactions

We initiated our investigation by quantifying microbial diversity within and between the dengue severity subgroups, employing alpha and beta diversity analyses to positional shifts linked to disease progression. The Shannon and Simpson diversity indices were calculated for alpha diversity for species richness and evenness, respectively, showing no significant differences across the mild, moderate, and severe patients (Fig 2A-I, II). In contrast, beta diversity analysis using the Bray-Curtis dissimilarity index reveals notable differences in the microbial community composition between the severity groups. This variation shows statistical significance PERMANOVA (p=0.02), indicating significant divergence in the microbial profiles among the clinical subgroups. Principal Coordinates Analysis (PCoA) supported this finding, displaying distinct clustering patterns reflective of the differential disease severity (S2 File) (Fig 2A-III).

To refine this analysis of microbial heterogeneity more deeply a filter on species present in at least 30% of samples and relative abundance threshold of more than 0.1% was applied. Taxonomic profiling at the phylum, genus, and species

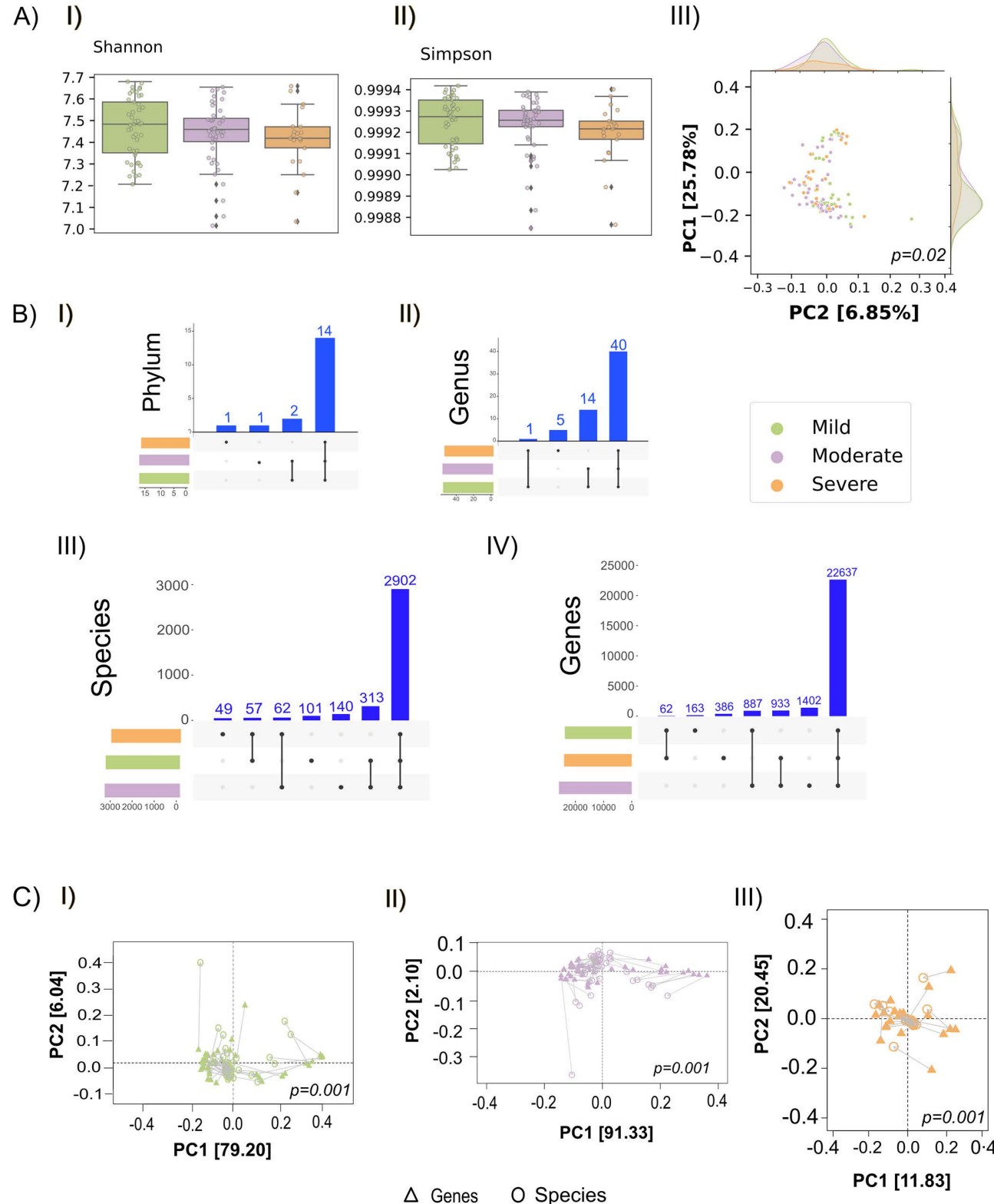

**Fig 2. Diversity driven functional profiling of host-TAMs interactions. A)** Box plot reflects the alpha diversity, specifically, **I)** Shannon, **II)** Simpson, which is measuring within sample species richness and evenness, respectively, and **III)** Beta diversity compares microbial composition between

samples, highlighting community dissimilarity. It is visualized using Principal Coordinates Analysis (PCoA) based on the distance matrices like Bray-Curtis. **B)** The total number of distinct phylum identified within each severity group (mild, moderate, and severe) reflects in the **B (I)**, **(II)** represents the number of genera present in each group, offering a more detailed view of the microbial composition changes. Total number of TAMs, genes and their distribution throughout the group is shown in **(III) & (IV)**. **C)** The Procrustes analysis visualizes the concordance between microbial and gene expression datasets by superimposing their ordination plots across **I)** Mild, **II)** Moderate, and **III)** Severe.

levels uncovered group-specific microbial patterns. Using a relative abundance threshold of more than 0.1%, we found 14 phyla common to all the severity groups. Looking at the different taxa, notable disparities in relative abundance across the severity spectrum was observed. The phylum Proteobacteria showed a comparative abundance in mild (38.7%) and moderate (40.6%) patients, while a noticeable decline was observed in severe patients (33.26%). *Firmicutes*, a major component of commensal microbiota, exhibited an increasing gradient with severity (38.3%, 40.4%, and 50.2%, respectively), potentially reflecting a compensatory microbial response or niche expansion under inflammatory stress. However, some phyla were uniquely associated within the specific subgroups. For, e.g., *Planctomycetes* appeared only in the severe patients, while *Deinococcus-Thermus* was exclusive to the moderate but absent in the mild. Additionally, *Thermotogae* and *Uroviricota* were shared between the moderate and severe patients (Fig 2B-I). We identified a total of 55, 54, and 46 genera in the mild, moderate, and severe patients, respectively (S3 File). For comprehensive understanding, we have included average count of dengue virus as well as all the microbial species, including the detected viral and archaeal reads as S7 File. Several genera were uniquely present or shared among specific groups, suggesting potential taxonomic markers associated with the disease severity. Genus-level distribution showed no presence of unique genera in the mild and moderate but five genera namely, Cupriavidus, Curtobacterium, Fischerella, Methylobacterium and Scytonema were found to be uniquely present in the severe patients (Fig 2B-II).

At the species level, a total of approximately 5,500 TAMs were initially identified across all samples. After applying filters as discussed earlier, refinement resulted in the detection of 3,373 species in the mild group, 3,417 in the moderate group, and 3,070 species in the severe group. Of these, 101 species were unique to mild cases, 140 to moderate, and 49 to severe. Shared species included 57 between mild and severe, 313 between mild and moderate, and 62 between moderate and severe. Notably, a core microbiome of 2,902 species was consistently present across all the severity groups, indicating a stable microbial population. Parallel analysis of host transcriptomes expressed genes across the cohort, highlighted a total of 23749, 25859, and 24018 biotypes (mRNA and lncRNA) for the mild, moderate, and severe, respectively. Among the 22,637 genes being common, 163 were found to be uniquely present in the mild, 1,402 in the moderate, and 386 in the severe (Fig 2B-IV). Shared host genes included 887 between mild and moderate, 62 between mild and severe, and 933 between the moderate and severe. Further, to evaluate the overall association between the host gene expression profiles and TAMs, we performed procrustes analysis on the filtered data. This analysis reveals the concordance between host gene expression and expressed microbial community dynamics with a correlation value of 1.00 and sum of square ($m^2$) of 0, indicating a perfect and significant agreement between the two configurations (Fig 2C, I-III).

### Severity-specific host-microbe interactions reveal divergent immune-metabolic pathways in dengue via SCCA

Our initial findings impelled us to investigate the potential association between the host genes and the TAMs. To explore these associations, we applied SCCA, a dimensionality reduction technique for integrated host-microbial transcriptomic data on the filtered dataset, across all cohorts. Hyperparameter tuning yielded optimal sparsity penalties of 0.23 for mild, 0.1 for moderate, and 0.17 for severe cases. Using these parameters, the SCCA model was fitted to identify subsets of correlated host genes and TAMs, referred to as SCCA components. Each component contains non-zero canonical loadings, representing joint variation between TAMs (V) and host genes (U). These components are computed iteratively, ensuring they remain uncorrelated across the iterations with the 300 possible iterations. Specifically, we identified all

correlated components per subgroup: 45 components in mild (U1-V1 to U45-V45), 46 components in moderate (U1-V1 to U46-V46), and 21 components in severe patients (U1-V1 to U21-V21). We subsequently selected the top correlated gene (U) and TAMs (V) components for each severity group. Categorically, we chose U1-V1 ($r^2 = 0.91$, p = 0.001) for mild, U1-V1 ($r^2 = 0.96$, p = 0.001) for moderate, and U10-V10 ($r^2 = 0.95$, p = 0.001) for severe (S4 File). These top correlated components, visualized in Fig 3A, I-III, exhibit strong positive associations.

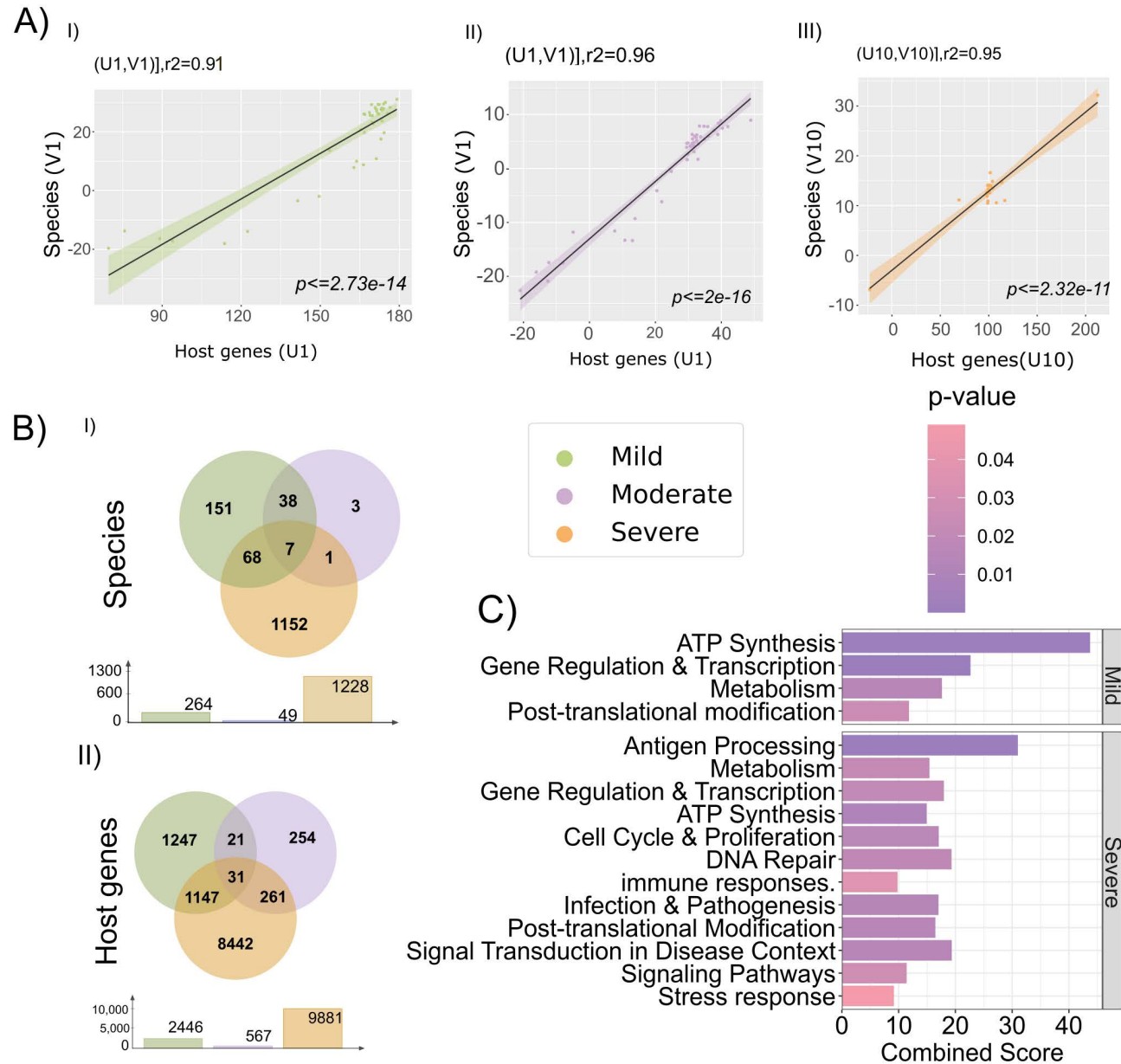

**Fig 3. Biological pathways modulation by the host genes through microbe interaction. A)** The correlation coefficients derived from SCCA are displayed in the linear model graphic. The direction shows the positive correlation between host specific genes and TAMs for, **I)** Mild, **II)** Moderate, and **III)** Severe. **B)** Venn diagram represents, **I)** species and, **II)** host genes overlap, based on the maximum canonical correlations group-wise. **C)** Represents the broader categories of significant pathways for the mild and severe.

Following SCCA analysis, the number of retained TAMs was 264 in the mild, 49 in the moderate, and 1,228 in the severe, while the corresponding number of genes was reduced to 2,446, 567, and 9,881, respectively. Among the TAMs, 151 were unique to the mild, 3 to the moderate, and 1,152 to the severe. Overlapping TAMs included 38 shared between mild and moderate, one between moderate and severe, 68 between mild and severe, and seven were common to all the three groups Fig 3B-I. Similarly, sets of genes were uniquely enriched in each severity group, with 1,247 genes in mild, 254 in moderate, and 8,442 in severe. Overlapping gene signatures included 21 genes shared between mild and moderate, 1,147 between mild and severe, 261 between moderate and severe, and 31 genes common to all the three (Fig 3B-II).

To investigate host biological pathways linked to the disease severity, we performed pathway enrichment analysis using Enrichr [19] [20] on the highly correlated and significant host genes (U) derived from the SCCA components: U1 for mild and moderate cases, and U10 for severe cases. The background genes for the enrichment analysis consisted of the full set of host genes retained after 30% filtering step, along with their corresponding U component values (U1 for mild and moderate, U10 for severe). Enrichment was carried out using the reactome pathway database, and statistical significance was evaluated using the Fisher's exact test. This analysis identified 1,450 pathways in the mild group, 581 in the moderate group, and 1,803 in the severe patients. Among these, based on the adjusted p-value <= 0.05 value, only 12 in the mild, and 199 pathways were significantly enriched in the severe patients, while no significant pathways were observed in the moderate (S5 File). To reduce noise and enhance the biological relevance of our analysis, we applied a filtering criterion to exclude significant pathways that contained fewer than 25 or more than 300 genes. This decision was based on the hypothesis that pathways with fewer than 25 genes may reflect incidental presence rather than functional involvement, while those with more than 300 genes may be too broad, diluting specific biological signals and potentially contributing to non-specific associations.

This resulted in 9 pathways including 624 host genes in the mild and 150 pathways including 2613 host genes in the severe. To investigate the functional relevance of all the significant pathways, we grouped similar pathways into broader functional categories. This process resulted in 4 distinct categories in the mild and 20 in the severe (S5 File). We identified a subset of 12 key pathways in the severe, along with four distinct pathway categories specific to the mild. The mild encompassed broader functional categories such as ATP Synthesis, Gene Regulation and Transcription, Metabolism, and Post-translational Modifications. In severe patients, there is broad enrichment of pathways related to Antigen Processing, Infection & Pathogenesis, DNA Repair, Immune Responses, Gene Regulation and Transcription, Post-translational Modifications, Disease-associated Signal Transduction, Signaling Pathways, Stress Response, Cell Cycle and Proliferation, ATP Synthesis and Metabolism. Notably, ATP synthesis and Metabolism were shared between both the groups but exhibited higher enrichment in the mild patients (Fig 3C).

These findings suggest that energy production and metabolic functions are more actively sustained during the milder stages of infection, while in severe cases, these processes may be down regulated due to the increased immune dysregulation and cellular stress. Collectively, these pathways indicate dynamic and responsive host efforts to detect, respond to, and repair the damage caused by the severe dengue infection. Antigen processing and immune response pathways aim to clear the virus, while DNA repair and stress response mechanisms work to preserve cellular integrity. Signal transduction and pathogenesis related pathways coordinate these processes but may also contribute to immunopathology, leading to tissue damage and disease severity. Next, while the mild group exhibited targeted enrichment patterns, indicating some involvement in disease progression, the absence of significant pathways in the moderate suggests, it may represent a transitional state with limited or no contribution. Nonetheless, the severe patients demonstrated the most pronounced biologically relevant pathways, highlighting its strong correlation with underlying factors contributing to severity.

## LASSO based host-microbial transcriptomic integration reveals transcriptionally active microbes and host genes shaping severe dengue

SCCA analysis piques us to investigate one-to-one correlation analysis of host genes and TAMs via applying LASSO regression followed by the stability selection to identify reliable host-microbiome associations linked to the disease

severity. These models were fitted in a gene-wise manner, using the expression level of each host gene as the response variable and microbial taxa abundances as predictors. This approach reduces false positives by selecting features with high predictive value and 95% confidence interval (CI), ensuring biological relevance (Fig 4A). This analysis focused on the mild and severe groups, where significant and biologically relevant pathways were identified. For the Lasso association analysis, we used the TAMs selected from the SCCA results and host genes identified from the significant and biologically relevant pathways. In the mild group, 624 host genes and 264 TAMs were included, resulting in 164,736 possible associations. In the severe, the model incorporated 2,613 host genes and 1,228 microbial taxa, yielding a total of 3,208,764 potential associations. We then applied stability selection to identify robust associations.

Using this approach, we identified 235 stability-selected host gene-taxa associations in the mild and 16 in the severe. These associations involved 199 host genes and 55 microbial taxa in the mild, and 16 host genes and 15 microbial taxa in the severe (S6 File). Among these selected host genes, 6 and 16 pathways are associated in the mild and severe respectively. The pathways from mild include essential cellular processes which based on their function we merged in broader categories, such as ATP synthesis, gene regulation and transcription, metabolism, and protein modification. (S5 File) Outlines the detailed pathways corresponding to the broad functional categories. Together, these pathways form the molecular foundation for maintaining cellular homeostasis, adaptability, and response to physiological stress. Notably the broad pathways identified based on their function in severe includes immune response, infection and pathogenesis, antigen processing, and metabolism, highlighting the complex host response to infection and cellular stress. Table 2 outlines the detailed pathways corresponding to the broad functional categories. Together, these interconnected systems reflect the coordinated biological response to infection, balancing defense, repair, and metabolic adaptation (Fig 4 B-I, C-I).

Further, literature on these pathway's genes showed that, for mild instances, 7 host genes were related to the outcome of the disease, as well as 5 TAMs linked to those genes were also related to the disease, as demonstrated by the network. Similarly, for the severe, 16 strong pathways demonstrating 7 host gene connections with 7 TAMs showed clinical significance and possible roles in dengue severity and progression. Across both the mild and severe groups, we identified key microbes with strong associations to the host genes and visualized their interactions to illustrate patterns of host gene and microbe crosstalk (Fig 4B-II and 4C-II). Among the identified genes in mild, *ZNF441* stood out as a transcription factor involved in multiple infectious contexts, including the regulation of herpes simplex virus (HSV) and cancer progression. *ZNF441* was associated with *S. griseus*, a well-known antibiotic-producing bacterium. *SMURF2*, an E3 ubiquitin ligase involved in the TGF-β signaling pathway and known to assist in protein degradation critical to IFN-1 antiviral signaling, was linked with *P. apista*, a multidrug-resistant pathogen implicated in chronic lung infections. Further, *MRPL34*, a mitochondrial ribosomal protein that plays roles in apoptotic and survival pathways in infectious diseases, was associated with *M. magneticum*, a TAM being explored for its potential role in cancer therapy. Another mitochondrial gene, *MRPL34*, was also linked to *M. magneticum*, reinforcing this microbial relevance in host regulatory functions. *PRMT3*, an arginine methyltransferase involved in modulating viral and antiviral immune responses through protein methylation, was similarly associated with *M. magneticum*. Additionally, *EIF4EBP1*, a translational repressor of the mTORC1 signaling pathway that plays a role in adaptive immune regulation, was linked with *C. ureolyticus*, a pathogen known to impact gastrointestinal immune responses (Fig 4B-II).

In severe, *PIK3R1*, a critical gene involved in regulating lymphocyte function and IL-9 signaling, was found to be associated with *C. uterequi*, a TAM linked to urogenital infections. *PPME1*, known to play a role in IL-8 induction through the ERK pathway, showed a relationship with *A. ghanensis*, which possesses immunomodulatory capabilities. Another notable finding involved *TIMP2*, a gene with anti-inflammatory properties that inhibits matrix metalloproteinases to prevent tissue damage, correlating with *B. licheniform,* an opportunistic microbe identified in immunocompromised individuals. Other significant gene-TAMs interactions included *CERK*, a gene implicated in metabolic inflammation, associated with *C. cochlearium*; *TLR4*, an essential component of innate immune defense that detects pathogen-associated molecular patterns (PAMPs), linked with the *E. albertii* diarrhea causing TAM; *TOP1*, which is vital for the activation of immune

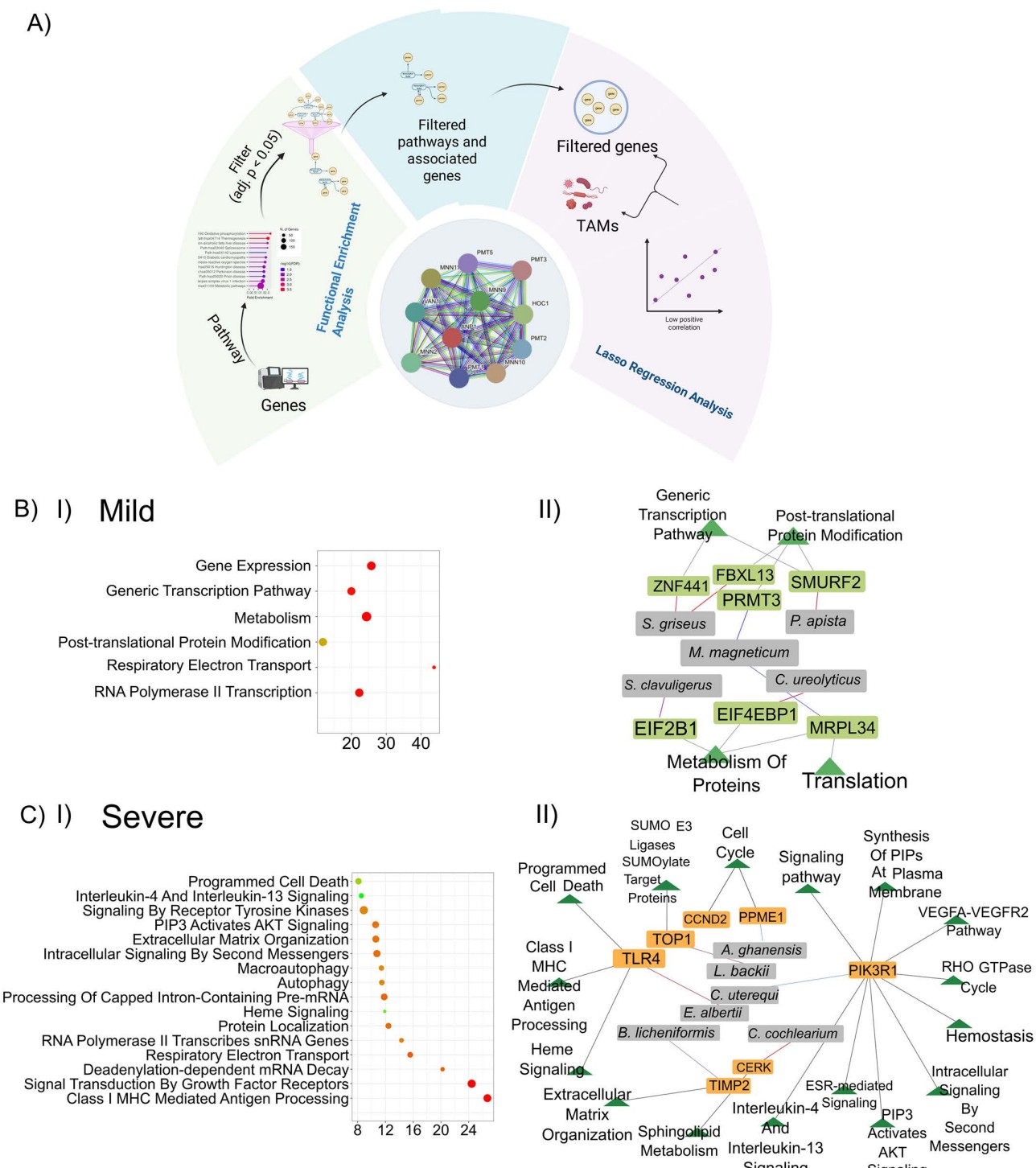

**Fig 4. Machine learning model identifies biological pathways affected in different disease severities. A)** Canonical steps for interpreting and selecting pathways and their associated genes, representing the Lasso regression model to uncover potential microbial contributors linked to these genes. (Figure created with Biorender.com. Devi, P. (2025) https://BioRender.com/qijg9l6) **B)** Mild, **C)** Severe: **I)** Pathways selected on the basis of biological relevance are shown though bubble plot, **II)** Network represent the selected genes and their associated pathways with TAMs based on significance level.

**Table 2. Enriched pathways identified from host gene analysis and their broad biological categories.**

| Mild | |
|---|---|
| **Major categories** | **Term** |
| ATP Synthesis | Respiratory Electron Transport |
| Gene Regulation & Transcription | RNA Polymerase II Transcription |
| | Generic Transcription Pathway |
| | Gene Expression |
| Metabolism | Metabolism |
| Post-translational modification | Post-translational Protein Modification |
| **Severe** | |
| Antigen Processing | Class I MHC Mediated Antigen Processing And Presentation |
| Signal Transduction in Disease Context | Diseases Of Signal Transduction By Growth Factor Receptors And Second Messengers |
| RNA Processing and Modification | Deadenylation-dependent mRNA Decay |
| | Processing Of Capped Intron-Containing Pre-mRNA |
| ATP Synthesis | Citric Acid (TCA) Cycle And Respiratory Electron Transport |
| Gene Regulation & Transcription | RNA Polymerase II Transcribes snRNA Genes |
| Intracellular Transport & Organelle Maintenance | Protein Localization |
| Autophagy | Autophagy |
| | Macroautophagy |
| Cellular Component Organization and Biogenesis | Extracellular Matrix Organization |
| Signaling Pathways | Intracellular Signaling By Second Messengers |
| | Heme Signaling |
| | PIP3 Activates AKT Signaling |
| | Signaling By Receptor Tyrosine Kinases |
| Immune response | Interleukin-4 And Interleukin-13 Signaling |
| Apoptosis | Programmed Cell Death |

response genes through RNA polymerase II, correlated with *L. backii* and *CCND2*, a cyclin that regulates intestinal immune homeostasis, associated with *L. pentosus*, a commensal bacterium recognized for its immune-modulatory roles (Fig 4C-II). These interactions revealed a strong enrichment of genes in severe, like PI3K-associated immune signaling pathways, including IL-4, and IL-13 signaling, estrogen receptor-mediated signaling, AKT, and VEGF-A and VEGFR2 pathways, which is responsible for the vascular permeability and plasma leakage in severe dengue (Fig 4C-I). Altogether, these findings highlight the presence of pro-inflammatory and immune-activating signatures in severe dengue, with host gene and transcriptionally active microbe associations, pointing to an interplay between the immune signaling disruption and opportunistic microbial activity, which may contribute to disease exacerbation. The severe patients display the most extensive and biologically enriched interactions, suggesting a complex yet dynamic host gene-microbiome reprogramming that amplifies disease severity.

## Discussion

Ample evidence underscores the pivotal role of host-microbe interactions modulating disease and highlighting potential therapeutic avenues. In our studies from the lab on COVID-19, we were the first to highlight the role of transcriptionally active microbial communities in shaping infectious disease severities and outcomes, particularly in patients infected with distinct SARS-CoV-2 variants of concern [15–17]. While Artificial Intelligence (AI) has gained significant traction in the early detection of chronic illnesses such as cancer and diabetes, its application in the diagnosis and prediction of infectious diseases remains relatively underutilized. This gap becomes even more evident when reviewing studies, where

AI-driven models have shown remarkable promise in other medical domains. Building on this, our current research integrates genomics-driven total RNA sequencing with a machine learning-based integrated host-microbial transcriptomic framework to decode the complex immuno-microbial landscape of dengue infection. This innovative approach not only bridges the gap between data-rich biology and AI but also opens new frontiers for precision diagnostics in infectious diseases.

Embarking on the microbial frontier of dengue pathogenesis, our study uncovered striking alterations in the blood microbial community through beta diversity analysis across the patient subgroups, spotlighting the nuance that may correlate with disease progression and severity. While a few reports, for instances Tan et al. highlighted no evidence for a consistent core blood microbiome in a multi-cohort study of 9,770 healthy humans over a period of time [21]. It is important to note that we have investigated only the confirmed Dengue virus infected hospitalized patients with differential disease severity based on the clinical data (mild, moderate, severe) and not healthy humans. In such pathological conditions, barrier dysfunction and systemic inflammation can facilitate transient microbial translocation into the bloodstream, which are transcriptionally active and interacts with the host immune system. Delving deeper into taxonomic trends, we noted a compelling reorganization of dominant bacterial phyla. *Proteobacteria* displayed a progressive increase in moderate in comparison to mild but a sudden decline in severe patients suggesting a potential breakdown in the protective microbial networks. In contrast, *Firmicutes* surged in abundance, potentially reflecting an expansion of microbial response under inflammatory stress [22, 23]. The concurrent depletion of *Bacteroidetes*, intimately linked with mucosal health and metabolic equilibrium, further reinforces the narrative of disrupted homeostasis [24]. Particularly, our interest was the emergence of a group of unique genera in the severe dengue cases. These genera tend to contain opportunistic pathogens such as *Cupriavidus* and *Curtobacterium*, typically rare in healthy individuals, appeared enriched, hinting at immune evasion in the severe patients. The detection of cyanobacteria such as *Fischerella* and *Scytonema*, known producers of hepatotoxic and neurotoxic compounds, raises provocative questions about their role in exacerbating systemic inflammation. Similarly, the presence of *Methylobacterium,* commonly associated with nosocomial infections, aligns with microbial signatures linked to compromised immunity [25, 26]. Together, these findings open a window into the dynamic expressed microbial re-landscaping and host genes that occurs during dengue infection and beckon deeper investigation into how these microbial shifts could inform severity (S2 Fig).

To unravel the intricate host gene and microbe crosstalk underlying dengue progression, we leveraged SCCA which enabled the identification of TAMs most strongly associated with host gene expression patterns. Interestingly, while both mild and severe dengue groups yielded distinct sets of enriched pathways, the severe group displayed a notably rich and complex molecular signature. This enrichment paints a picture of a heightened and multifaceted immune engagement in the severe cohort potentially representing both a response to viral burden and an indicator of systemic immune dysregulation. In contrast, the mild group exhibited a comparatively subdued pathway landscape, with sub-optimal immune and infection-related pathways. This apparent immunological silence may suggest a more contained host response or a less disruptive microbial presence during early or less severe infection stages.

Leveraging LASSO regression, we identified robust, severity-specific host gene and microbe associations, highlighting the utility of machine learning in unraveling complex biological interactions. Mild cohort associated with primarily immune regulation, like *ZNF441* and *S. griseus,* and *SMURF2* associated with *P. apista*, suggest a role for microbial partners in modulating antiviral and anti-inflammatory responses [27, 28]. Associations with *M. magneticum* and mitochondrial genes like *MRPL34* and *MRPS34* point to potential microbiome involvement in the host metabolic and apoptotic pathways during early infection stages [28]. In contrast, the severe group showed enrichment in pro-inflammatory and immune-activating pathways such as *PI3K*, IL8, IL-4, IL-13, ERK, and VEGF signaling play key roles in acute and chronic inflammation, contributing to diseases such as meningitis, pneumonia, sepsis, and urinary tract infections [29–32]. In contrast, their correlated TAMs are associated with chronic respiratory and intestinal infections too. Among the most relevant observations are the associations of *PIK3R1* and *C. uterequi*, *PPME1* and *A. ghanensis*, as well as *TIMP2* and *B. licheniformis*, all

implicating microbial influence on the immune regulation and tissue integrity [33–37]. These findings suggest that opportunistic microbial activity may exacerbate host immune dysregulation, contributing to the severity of dengue. Our study highlights the value of integrating host gene expression and microbiome data to uncover their participatory contributions to dengue. While some associations are disease-specific, others reveal that the same microbes can influence different host genes across disease severity and likewise, identical host pathways may be linked to distinct microbes.

## Methods

### Ethics statement

The study was designed in accordance with the Declaration of Helsinki and was approved by the institutional ethics committee of Council of Scientific & Industrial Research-Institute of Genomics and Integrative Biology, Delhi, India (Ref No: CSIR-IGIB/IHEC/2020–21/01). The patients/participants provided their written informed consent before participation in this study.

### Study design

This study was conducted at the CSIR-Institute of Genomics and Integrative Biology (IGIB), Delhi, India, using total RNA isolated from the blood sample collected by the paramedical team at the MAX Super Speciality Hospital, Delhi, on the very day when patients reported to the hospital. This includes 112 patients who tested positive for dengue during the peak dengue season in India (August-November 2022). Retrospective clinical data were collected by reviewing the electronic medical records of each patient. Based on the complete blood count (CBC) reports, the 112 NS1-positive dengue patients were classified into three subgroups. As per the World Health Organization (WHO) dengue classification and management guidelines for severe category, none of the patients from our study exhibited the symptoms of severe bleeding, organ failure, or abnormal liver parameters. Consequently, a separate category for severe dengue was not established. To ensure clarity, our patient subgroups are designated as Mild (no Thrombocytopenia or Leukopenia), Moderate (Leukopenia without Thrombocytopenia), and Severe (both Thrombocytopenia and Leukopenia) dengue cases. The mild group consisted of 45 patients with normal platelet and leukocyte counts. The moderate group included 46 patients with normal platelet counts but decreased leukocyte counts (Leukopenia). The Severe group comprised 21 patients with both low platelet counts (Thrombocytopenia) and decreased leukocyte counts (Leukopenia) [9].

### Isolation of transcriptomic material

RNA extraction from the blood samples was performed using the Qiagen QIAamp RNA Blood Mini Kit, following the manufacturer's protocol with specific modifications. The incubation and centrifugation time during the erythrocyte lysis step was reduced to five minutes, while an additional 2–3 minutes of incubation was introduced during all washing steps to enhance RNA purification. The purity of the extracted RNA was assessed using a NanoDrop Microvolume Spectrophotometer and further visualized by agarose gel electrophoresis. The purified total RNA was then stored at -80°C until thawed for RNA-seq library preparation [38].

### Library preparation and sequencing

RNA libraries were prepared using 250 ng of total RNA with the Illumina TruSeq Stranded Total RNA Library Prep Globin kit (Cat. No. 20020612). Both globin mRNA and ribosomal RNAs-highly abundant in whole blood were depleted during the initial steps. The resulting RNA fragments were reverse transcribed into first-strand cDNA using random primers, followed by second-strand synthesis with DNA polymerase I and RNase H. The double-stranded cDNA was purified using AMPure XP beads (Beckman Coulter, A63881), adenylated at the 3′ ends, and indexed for sequencing through PCR amplification.

Library quality was assessed via fragment size analysis using an Agilent 2100 Bioanalyzer, and concentrations were quantified with the Qubit dsDNA High Sensitivity Assay Kit (Thermo Fisher Scientific, Q32854). Libraries were normalized to 4 nM, pooled in equimolar ratios (24 samples per pool), and sequenced on the Illumina NextSeq 2000 platform using paired-end 2 × 151 bp reads at a final loading concentration of 650 pM [38].

## Preprocessing and filtering of host reads from meta-transcriptomic data

Basecalls generated by Illumina sequencing were converted into the standard compressed FASTQ file format using bcl-2fastq for downstream analysis. The quality of raw reads for each sample was assessed using FastQC [39], and adapter/low-quality sequences were removed with Trimmomatic v0.39 [40]. Reads were then mapped to the human reference genome (GRCh38) using HISAT2 [41] to quantify transcript abundance. Counts were extracted using DESeq2. The aligned human RNA reads were then separated using Samtools v1.17 [42, 43], generating human reference mapped, and those that could not map to the human reference are designated as "human-unmapped RNA-seq data" and processed separately, for non-canonical meta-transcriptomic analysis [38].

## Functional pathway enrichment analysis

Functional pathway of human mapped genes extracted from DESeq2 analysis was performed using the Enrichr web tool [19] utilizing pathway and term annotations from the Reactome database. Pathways with an adjusted p-value ≤ 0.05 were considered statistically significant. Second filter was applied on the basis of the genes present in the particular pathway, for ex: pathways containing fewer than 25 or more than 300 genes were excluded to avoid biases in the pathway. This filtering ensured the retention of pathways with biological relevance. The results were visualized in R using the ggplot2 package, where the combined score was plotted against the number of genes associated with each pathway. In the plots, adjusted p-values were represented by color gradients, and bubble size indicated gene counts present in the particular pathway.

## Taxonomic classification of TAMs

Extracted human-unmapped reads were taken for meta-transcriptomic analysis. Wherein taxonomic classification of microbial communities was performed using Kraken2 and Bracken2 [44, 45]. Kraken2, a k-mer-based taxonomic classifier, was used to assign microbial taxonomy by mapping reads to a genomic database containing bacterial, archaeal, and viral reference sequences (Minikraken v1) and kraken2-build was used to build the database [46, 47]. Since kraken2 does not assign every read to the species level, bracken2 (Bayesian Re-estimation of Abundance after Classification with Kraken2) was used to estimate species-level read abundance [38]. An in-house Python script was developed to combine Bracken output with lineage information and convert it into a BIOM file using Kraken-Biome, facilitating downstream analysis in R. Following quality control and filtering, meta-transcriptomic reads are taxonomically classified at various taxonomic levels, from phylum to species. Prevalence and abundance-based filtering ensured retention of taxa present above a defined relative abundance threshold in a specified percentage of samples. To account for the compositional nature of microbiome datasets, normalization was done using cumulative sum scaling (CSS) transformation [48]. This ensures that differences in sequencing depth do not bias downstream analyses. Alpha and beta diversity analyses were conducted in R (v4.2.0) [49] using vegan (v2.6-2) [50] and phyloseq (v1.40.0) [51]. Alpha diversity (Shannon and Simpson indices) was calculated using the estimate_richness function from phyloseq. Beta diversity was assessed using the Bray-Curtis dissimilarity matrix, generated via phyloseq::distance from the vegan package. Principal Coordinate Analysis (PCoA) was performed for visualization. The analyzed data was then exported as a CSV file for further plotting and visualization in Python (v3.6) [52] using Seaborn (v0.12.2) and Matplotlib (v3.1) [52], built on NumPy (v1.11.0) [53].

### Host-TAMs interactions and model fitting

SCCA is a statistical technique designed to project two datasets into a shared latent space, maximizing their correlation. By incorporating L1 (lasso) penalties, SCCA enhances feature selection, making it particularly well-suited for high-dimensional datasets [54–56]. For each disease cohort, SCCA is applied using the R package 'PMA', treating microbiome taxonomic composition and host gene expression as two sets of variables to be correlated [54]. Once the optimal sparsity parameters are determined using "CCA.permute", the SCCA model is fitted to extract subsets of correlated host genes and TAMs using function "CCA" and the output referred to as SCCA components. Each component contains non-zero canonical loadings, representing joint variation between the TAMs and host genes. These components are computed iteratively, ensuring they remain uncorrelated across iterations. Sparse CCA is a useful statistical tool to detect latent information with sparse datasets. Correlation was done to the scores calculated for the held-out samples in order to evaluate the actual strength of the correlation and its significance. The Benjamini-Hochberg (FDR) was used to adjust the p-values for multiple hypothesis testing within each disease severity subgroups. Significant components were identified at FDR<= 0.05. This examined the relevance of a set of host genes that were connected with a set of bacteria at the component level, as opposed to the significance of the individual characteristics chosen, which is contingent upon the degree of sparsity penalization.

### Regularized regression using LASSO for biomarker discovery

To enhance the reliability of our findings and reduce model instability, we applied LASSO (Least Absolute Shrinkage and Selection Operator) regression [57]. This was followed by stability selection to identify specific host genes, and transcriptionally active microbiome associations, potentially contributing to the differential disease severity. This regularization method adds an L1 penalty to regression coefficients, driving irrelevant features to zero and retaining those with predictive value. This technique identifies variables selected with a cutoff of 0.6 to define robust associations with 95% confidence intervals, minimizing false positives while preserving biological relevance.

### Procrustes analysis for integrated host-microbial transcriptomic data integration

Procrustes analysis performed using "protest()" function with 999 possible combinations and shown in Fig 2C, I-III (Mild, Moderate, Severe), the concordance between the transcribed microbial and human host gene expression datasets by superimposing their ordination plots [58]. Each pair of points (TAMs and host genes) is connected by a line, where shorter lines indicate a stronger agreement between the datasets. The overall fit is quantified using the Procrustes sum of squares ($M^2$) and its associated p-value, which reflect the strength and significance of the correlation.

### Quantification and statistical analysis

To compare clinical significance across the subgroups, continuous numerical variables were analyzed using the Mann-Whitney U test. Microbial count normalization was performed using the CSS (Cumulative Sum Scaling) method, a median-like quantile normalization approach, from the R package metagenomeSeq. Alpha diversity significance (Shannon and Simpson indices) was assessed using the kruskal wallis test, while beta diversity significance was determined via permutational multivariate ANOVA (PERMANOVA (Significance p-value<0.02)). All statistical analyses and data visualization were performed in R v4.2.0 using the following packages: phyloseq, vegan, tidyverse, dplyr, Adonis2, ggplot2, ggpubr, circlize, and cytoscape. Additionally, Python (v3.6) was used for visualization with Seaborn (0.12.2) and Matplotlib (3.1), built on NumPy (1.11.0).

### Conclusion

Our integrative analysis reveals that the transcriptionally active microbial profiles and host gene signatures closely align with clinical indicators of dengue severity. These results point to a progressive remodeling of host-microbiome networks with increasing disease severity, marked by more extensive and biologically enriched interactions in the severe group. The moderate group appears to represent a transitional phase, with reduced host genes and TAMs association, whereas

the mild group retains a subset of functional interactions, contributing modestly to disease severity. The severe groups of enriched immune and inflammatory pathways correspond with clinical findings such as lymphocytosis, thrombocytopenia, and elevated liver enzymes hallmarks of systemic immune activation. Using LASSO regression, we identified severity-specific associations between host genes and microbes, revealing a shift from immune-regulatory interactions in mild dengue to pro-inflammatory and immune-disruptive networks in severe cases. Key associations-such as *ZNF441* and *S. griseus* in mild and *PPME1* and *A. ghanensis* causing immune dysregulation in severe cases highlight the dynamic role of microbes in modulating host responses. These findings underscore the dual role of host genes and TAMs as modulators and markers of disease progression.

## Supporting information

**S1 Fig. Percentage composition of human-mapped and human-unmapped reads, with microbial content derived from human-unmapped reads fraction.**
(TIFF)

**S2 Fig. Illustrates significant host gene-TAMs associations and their functional roles in modulating immune and inflammatory pathways contributing to severe dengue. (Figure created with Biorender.com. Devi, P. (2025) https://BioRender.com/neii33a).**
(TIFF)

**S1 Table. Clinical distribution across dengue disease severities.**
(XLSX)

**S1 File. Distribution of total reads with human mapped reads and unmapped microbial reads for all samples from the RNA-Seq data set.**
(XLSX)

**S2 File. Diversity classification of transcriptionally active microbes (Alpha diversity and Beta Diversity).**
(XLSX)

**S3 File. Phylum and genus level taxonomic communities across dengue severity groups.**
(XLSX)

**S4 File. Sparse Canonical Correlation Analysis along with their canonical variates V (microbes) and U (genes).**
(XLSX)

**S5 File. Functional pathways of mild Dengue patients using reactome pathway database.**
(XLSX)

**S6 File. Stability selection of host genes and microbes using LASSO for mild.**
(XLSX)

**S7 File. Comprehensive list of average count of dengue virus and all the microbial species, including the detected viral and archaeal reads.**
(XLSX)

## Acknowledgments

Authors would like to acknowledge all the dengue patients who participated in the study. Authors duly acknowledge the help and support from Dr. Bharti Kumari as lab manager, towards facilitation and coordination with the funders. Authors further recognize P.L. Bhatt for his contribution in providing samples. Authors acknowledge the support of Anil Kumar and Nisha Rawat for their assistance with sample transport and management.

## Author contributions

**Conceptualization:** Rajesh Pandey.

**Data curation:** Pallawi Kumari, Priti Devi, Tav Pritesh Sethi.

**Formal analysis:** Pallawi Kumari.

**Funding acquisition:** Rajesh Pandey.

**Investigation:** Pallawi Kumari, Priti Devi, Rajesh Pandey.

**Methodology:** Pallawi Kumari, Priti Devi, Basudha Banerjee.

**Project administration:** Tav Pritesh Sethi, Rajesh Pandey.

**Resources:** Bansidhar Tarai, Sandeep Budhiraja, Rajesh Pandey.

**Supervision:** Tav Pritesh Sethi, Rajesh Pandey.

**Visualization:** Pallawi Kumari, Priti Devi, Basudha Banerjee.

**Writing – original draft:** Pallawi Kumari, Priti Devi, Basudha Banerjee.

**Writing – review & editing:** Tav Pritesh Sethi, Rajesh Pandey.

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
