## [Decision Letter · Decision Letter 0]

9 Oct 2025

Functional dynamics between resident transcriptionally active microbes (TAMs) and host genes underlie Dengue severity

Dear Dr. Pandey,

Thank you for submitting your manuscript to PLOS Neglected Tropical Diseases. After careful consideration, we feel that it has merit but does not fully meet PLOS Neglected Tropical Diseases's publication criteria as it currently stands. Therefore, we invite you to submit a revised version of the manuscript that addresses the points raised during the review process.

Please submit your revised manuscript within 60 days Dec 08 2025 11:59PM. If you will need more time than this to complete your revisions, please reply to this message or contact the journal office at plosntds@plos.org. Please include the following items when submitting your revised manuscript:

We look forward to receiving your revised manuscript.

Kind regards,

Gyaneshwer Chaubey, PhD

Guest Editor

David Safronetz

Section Editor

Shaden Kamhawi

co-Editor-in-Chief

Paul Brindley

co-Editor-in-Chief

**Journal Requirements:**

At this stage, the following Authors/Authors require contributions: Pallawi Kumari, Priti Devi, Basudha Banerjee, Bansidhar Tarai, Sandeep Budhiraja, Tav Pritesh Sethi, and Rajesh Pandey. Please ensure that the full contributions of each author are acknowledged in the "Add/Edit/Remove Authors" section of our submission form.

2) Some material included in your submission may be copyrighted. According to PLOSu2019s copyright policy, authors who use figures or other material (e.g., graphics, clipart, maps) from another author or copyright holder must demonstrate or obtain permission to publish this material under the Creative Commons Attribution 4.0 International (CC BY 4.0) License used by PLOS journals. Please closely review the details of PLOSu2019s copyright requirements here: PLOS Licenses and Copyright. If you need to request permissions from a copyright holder, you may use PLOS's Copyright Content Permission form.

Potential Copyright Issues:

- Figures 1, 4, and 5. Please confirm whether you drew the images / clip-art within the figure panels by hand. If you did not draw the images, please provide (a) a link to the source of the images or icons and their license / terms of use; or (b) written permission from the copyright holder to publish the images or icons under our CC BY 4.0 license. Alternatively, you may replace the images with open source alternatives. See these open source resources you may use to replace images / clip-art:

3) Please ensure that the funders and grant numbers match between the Financial Disclosure field and the Funding Information tab in your submission form. Note that the funders must be provided in the same order in both places as well.

State what role the funders took in the study. If the funders had no role in your study, please state: "The funders had no role in study design, data collection and analysis, decision to publish, or preparation of the manuscript.".

**Reviewers' Comments:**

Reviewer's Responses to Questions

**Key Review Criteria Required for Acceptance?**

**Methods:**

-Are the objectives of the study clearly articulated with a clear testable hypothesis stated?

-Is the study design appropriate to address the stated objectives?

-Is the population clearly described and appropriate for the hypothesis being tested?

-Is the sample size sufficient to ensure adequate power to address the hypothesis being tested?

-Were correct statistical analysis used to support conclusions?

-Are there concerns about ethical or regulatory requirements being met?

Reviewer #1: Methods  Objectives & Hypothesis: Yes. The objectives are clearly stated, focusing on host–microbe interactions in dengue severity, with a testable hypothesis that transcriptionally active microbes (TAMs) influence host gene expression .

Study Design: Appropriate. Dual RNA-Seq with machine-learning integration is well suited to address the hypothesis.

Population: The patient cohort (112 NS1-positive dengue cases, stratified by severity) is clearly described and appropriate .

Sample Size: Adequate to detect relevant associations and supported by robust statistical analyzes (SCCA, LASSO).

Statistical Analyses: Correct and well described, including normalization, diversity metrics, sparse CCA, and LASSO regression .

Ethics/Regulatory: All ethical approvals and informed consents are clearly documented.

Reviewer #2: The Methods section requires substantial improvement because it is missing several key details, including information on pathway analysis, enrichment, gene expression analysis, etc.

The human reference assembly is a specific haplotype, and not all sequences will map to it. For validation, BLAST should be performed on the unmapped data. Did the authors also try the Telomere-to-Telomere (T2T), the first complete, gapless sequence of a human genome?

Although the database tested for taxonomy classification includes virus, it's surprising that viral reads have not been reported. Were any dengue virus reads identified?

kraken2 DB name and version missing

Details needed for QC and filtering of metatrasnscriptomic reads

Is the normalization of the relative abundance obtained from Bracken relevant? This normalization may have been performed prior to the taxonomic or relative abundance analysis.

Despite its strengths, Lasso regression has some limitations. How do the authors address the bias caused by regularization and the issue of highly correlated features? Instead of distributing weights among correlated features, Lasso may arbitrarily select one and ignore the others, potentially missing useful information.

**Results:**

-Does the analysis presented match the analysis plan?

-Are the results clearly and completely presented?

-Are the figures (Tables, Images) of sufficient quality for clarity?

Reviewer #1: Analysis vs. Plan: Analyzes match the described methodology, including multi-omics integration and pathway enrichment.

Clarity & Completeness: Results are thoroughly presented with detailed tables, figures, and supplementary files.

Figures/Tables: High quality and sufficient for clarity.

Reviewer #2: Abbreviations should be detailed the first time they are mentioned in the text.

Very low levels of microbial reads mapped to the Kraken2 database. This clearly highlights the low level of mapping.

The read numbers don't correspond to those present in the S1 file.

The PCoA graph is too small to follow, but there does not seem to be any clustering pattern. The Shannon and Simpson indices of the sample are very similar, highlighting that there is no significant difference within the samples.

Figure 2 has way too many subgraphs, and they are too small with small text, making it difficult to follow up on.

It’s incorrect to say that there is a non-human fraction. As the mapping shows, <20% of the unmapped reads mapped to the microbial DB.

**Conclusions:**

-Are the conclusions supported by the data presented?

-Are the limitations of analysis clearly described?

-Do the authors discuss how these data can be helpful to advance our understanding of the topic under study?

-Is public health relevance addressed?

Reviewer #1: Support by Data: Conclusions are fully supported by the presented evidence, linking TAM–host interactions to dengue severity.

Limitations: Limitations such as transitional features of the moderate group are acknowledged.

Advancement of Knowledge: The study advances understanding of host–microbiome dynamics in dengue and demonstrates the utility of machine-learning–based multi-omics.

Public Health Relevance: Yes. Findings have clear implications for severity prediction and potential therapeutic targets.

Reviewer #2: (No Response)

**Editorial and Data Presentation Modifications?**

Reviewer #1: No essential changes required. A brief proof-reading for minor typographical consistency (eg, spacing in figure legends) would enhance clarity.

Reviewer #2: Why is the study considered multi-omics when only the transcriptomics dataset has been analyzed?

Figure 5 is unnecessary; it's more suited for a presentation or poster rather than a scientific publication.

Without line numbers, it is difficult to provide specific comments for the corresponding lines.

**Summary and General Comments:**

Reviewer #1: (No Response)

Reviewer #2: No Dengue virus reads have been reported. Although the Krakten2 DB did include a virus. Could the author comment on this?

It is interesting to see that several microbial reads have been reported. Human blood is traditionally considered sterile; however, recent studies have raised questions about the existence of a blood microbiome in healthy individuals.

A recent study from Tan et al. (https://www.nature.com/articles/s41564-023-01350-w) suggests that there does not exist a consistent core microbiome endogenous to human blood—instead, a transient and sporadic translocation of commensal microbes from other body sites into the bloodstream. No species were detected in 84% of individuals (n = 9770), while the remainder had a median of only one species.

PLOS authors have the option to publish the peer review history of their article (what does this mean? ). If published, this will include your full peer review and any attached files.

**Do you want your identity to be public for this peer review?** For information about this choice, including consent withdrawal, please see our Privacy Policy .

Reviewer #1: No

Reviewer #2: No

**Figure resubmission:**

**Reproducibility:**



---

## [Decision Letter · Decision Letter 1]

25 Nov 2025

Response to Reviewers
Revised Manuscript with Track Changes
Manuscript

Shaden Kamhawi

co-Editor-in-Chief

Paul Brindley

co-Editor-in-Chief

**Journal Requirements:**

At this stage, the following Authors/Authors require contributions: Pallawi Kumari, Priti Devi, Basudha Banerjee, Bansidhar Tarai, Sandeep Budhiraja, Tav Pritesh Sethi, and Rajesh Pandey. Please ensure that the full contributions of each author are acknowledged in the "Add/Edit/Remove Authors" section of our submission form.

**Reviewers' comments:**

Reviewer's Responses to Questions

**Key Review Criteria Required for Acceptance?**

**Methods**

-Are the objectives of the study clearly articulated with a clear testable hypothesis stated?

-Is the study design appropriate to address the stated objectives?

-Is the population clearly described and appropriate for the hypothesis being tested?

-Is the sample size sufficient to ensure adequate power to address the hypothesis being tested?

-Were correct statistical analysis used to support conclusions?

-Are there concerns about ethical or regulatory requirements being met?

Reviewer #1: 1-Are the objectives of the study clearly articulated with a clear testable hypothesis stated?:

-Yes. The writers spell out exactly what they’re looking into. How active microbes affect host genes depending on how bad dengue gets. Though there’s no single standard hypothesis being tested, there’s a clear question still the goals make sense. They line up with how the study was set up.

2-Is the study design appropriate to address the stated objectives?

- Yes. Looking at 112 people’s blood with dual RNA-Seq gives a firm ground to check how human and microbe genes act together. Instead of just combining a lot of data, they mix it smartly using sparse CCA along with LASSO, in ordr to spot complex links between host and microbes. That setup works well for finding patterns tied to disease levels. Also, splitting folks into mild, medium, or serious sickness based on blood signs makes sense here.

3-Is the population clearly described and appropriate for the hypothesis being tested?

- Yes. The group traits like symptoms, background info, and how serious their illness was, are spelled out in an easy manner. Pulling data from NS1-confirmed cases at one hospital over a set duration is keeping things uniform. That setup works well when looking into how the body reacts to the virus during dengue infection.

4-Is the sample size sufficient to ensure adequate power to address the hypothesis being tested?

- Yes. Having 112 transcriptomes is plenty for a complex combined analysis. Since the team later uses filters, reduces variables, then applies shrinkage methods, this number fits well. It is normally very common in multi-layered biological studies. The split by illness level "45 mild cases, 46 moderate and 21 serious cases" covers most patient types fairly. Even though fewer people had severe symptoms, that matches how often it happens in hospitals, so results still hold up.

5- Were correct statistical analysis used to support conclusions?

- Yes. The authors used sufficient tools for diversity checks, Community compostion differences for selecting robust pairwise assosciations viz. alpha plus beta, PERMANOVA and LASSO regression for selecting robust pairwise associations. These widely-used methods help combine different types of omics data. Most analyses hold up well and support the takeaway message of this manuscript. During review, minor issues came up like which version of Kraken2 DB was used, how microbe levels were adjusted, or how correlated features in LASSO were managed? But those points got cleared up. Even with these small gaps, the core analysis still stands solid.

6- Are there concerns about ethical or regulatory requirements being met?

- I dont see any issue here. Approval from the ethics board is given, complete with official codes plus proof that patients agreed. The research works with de-identified medical samples, while the team follows both university rules and journal policies. The raw sequence files are uploaded to an open-access archive as per the guidelines.

Reviewer #2: Thanks for updating the methods. It provides good clarity now.

**Results**

-Does the analysis presented match the analysis plan?

-Are the results clearly and completely presented?

-Are the figures (Tables, Images) of sufficient quality for clarity?

Reviewer #1: 1--Does the analysis presented match the analysis plan?

-Yes. The findings in the paper line up well with the steps laid out. First, they check data quality for both human and microbe sequences. Then they look at variety among microbes. After that, they combine host and microbe gene activity using sparse CCA along with LASSO modeling. Next comes an examination of enriched biological pathways. That order fits exactly what was said before in the methods section. Comparisons based on illness level, "light, medium, strong", fit the grouping explained earlier. Moving from basic makeup checks to linked patterns, then figuring out possible functions, makes sense and ties back to what the research aimed to do.

2--Are the results clearly and completely presented?

- Yes, overall it is well presented. Informations are sorted neatly and covers everything needed. Summary stats make sense, plus there are solid checks on variety and gene mapping. Multi-omics bits are clear, along with path activity links. Links between TAMs and host genes show up plainly. Story flows is fine to show how things might work. Extra files help support the core reselts and add clarity to them.

In the first riview, there were questions about how microbial reads were counted, how dengue virus data was matched, also why certain parts were picked as key in SCCA. Those points are addressed well in the updated version. Thanks to these modifications, takeaways now match how the data actually link up, making the whole study more grounded.

3--Are the figures (Tables, Images) of sufficient quality for clarity?

- Yes, but there aare still a few small issues. However it is not big and nothing major in it to contradict the publication. Most images show results clearly, good enough quality overall. Important parts like diversity charts, Procrustes maps, or SCCA visuals make good sense overall. Labels are mentioned well and they aren't confusing. After edits, figures got easier to grasp. For example: text is bigger now, colors pop better.

Some multi-part images, like the one showing species variety and groups, are a bit packed. Yet they are clear enough for publishing. Even though that diagram meant to explain ideas was called more "visual" before, it works fine to show how analysis steps fit together without confusing things. All in all, these tables and charts back up the findings well.

Reviewer #2: see below

**Conclusions**

-Are the conclusions supported by the data presented?

-Are the limitations of analysis clearly described?

-Do the authors discuss how these data can be helpful to advance our understanding of the topic under study?

-Is public health relevance addressed?

Reviewer #1: 1--Are the conclusions supported by the data presented?

-Yes. The findings reported match the results observed during the research. The paper shows clear indications that host gene activity and active microbes differ based on illness level in individuals with dengue. Using combined data methods like sparse canonical correlation and LASSO models reveals linked human genes and bacterial types varying by disease stage, while the explanation fits closely with the recorded outcomes. The authors toned down causal wording during revisions; as a result, findings now match correlation patterns instead of implying direct mechanisms. In general, what is concluded lines up with the data analysis.

2- Are the limitations of analysis clearly described?

- Yes. The researchers note key limits such as; issues tied to weak microbial signals in blood, the study’s snapshot design, possible lab-related variations, along with difficulties turning machine learning patterns into biological insights. Still, these points are addressed reasonably well. The interpretation reflects a grounded sense of what dual RNA-Seq is able to show. Although some technical hurdles persist such as detecting tiny microbial traces or a small number of severe cases, the authors openly acknowledges them.

3-Do the authors discuss how these data can be helpful to advance our understanding of the topic under study?

- Yes. The updated version better explains how results fit into current knowledge about dengue development. Authors stress examining host–microbiome links in active viral illness, while showing that certain host–microbe patterns may guide follow-up experiments, detection tools, or treatment ideas. By combining live microbial activity with human gene data, the work fills a key void i.e. shifting focus away from narrow, single-factor approaches. These points clearly boost the paper’s value.

-Is public health relevance addressed?

- Yes. The researchers clearly highlight the health significance of their work. Dengue still heavily affects medical services where it’s common. Therefore, knowing what drives severe cases matters for sorting patients, using resources wisely, and handling treatment. Their paper connects lab results to real-world uses like spotting high-risk individuals sooner, detecting signs of immune imbalance, or uncovering microbes that worsen illness. They explain how studying human-microbe relationships in dengue could lead to better outcome predictions and inform targeted therapies down the line.

Reviewer #2: see below

**Editorial and Data Presentation Modifications?**

Reviewer #1: The manuscript is well-structured. However, small tweaks in formatting or wording could improve how clear it reads such as:

1- Some multi-panel visuals, especially the one on species variety and classification, are still packed tightly, making text hard to read. Bigger fonts plus larger sections could help clarity a lot.

2- A few abbreviations like TLC or SGOT are used before explaining what they mean. Since AST is also introduced without definition, spelling out each one early would help a wider audience grasp them faster.

Overall Recommendation: Accept

Reviewer #2: Figure 5 can be moved to the supplementary information.

**Summary and General Comments**

Reviewer #1: he manuscript is well constructed, addresses an important knowledge gap, and offers meaningful advances in our understanding of dengue pathophysiology. With only minor editorial refinements suggested, I believe it now meets the standards for publication in PLOS Neglected Tropical Diseases. I am pleased to recommend acceptance after minor editorial adjustments, and I congratulate the authors on a strong and impactful piece of work.

Reviewer #2: I thank the authors for addressing most of the points. Could they also please address the following point?

1. Thank you for updating the Dengue virus read counts. However, viral read counts are 4-5 times higher in mild patients than in those with moderate or severe cases. Could they discuss in the paper why this discrepancy exists?

2. Please modify the language and remove multi-omics.

3. Please mention and discuss the blood microbiota, especially the study by Tan et al., which has a huge cohort of nearly 10,000.

PLOS authors have the option to publish the peer review history of their article (what does this mean? ). If published, this will include your full peer review and any attached files.

**Do you want your identity to be public for this peer review?** For information about this choice, including consent withdrawal, please see our Privacy Policy .

Reviewer #1: No

Reviewer #2: No

**Figure resubmission:**
---

## [Editor Report · Decision Letter 2]

8 Dec 2025

Dear Dr Pandey,

We are pleased to inform you that your manuscript 'Functional dynamics between resident transcriptionally active microbes (TAMs) and host genes underlie Dengue severity' has been provisionally accepted for publication in PLOS Neglected Tropical Diseases.

Best regards,

Gyaneshwer Chaubey, PhD

Guest Editor

David Safronetz

Section Editor

Shaden Kamhawi

co-Editor-in-Chief

Paul Brindley

co-Editor-in-Chief

---

## [Editor Report · Acceptance letter]

Dear Dr. Pandey,

We are delighted to inform you that your manuscript, "Functional dynamics between resident transcriptionally active microbes (TAMs) and host genes underlie Dengue severity," has been formally accepted for publication in PLOS Neglected Tropical Diseases.

Best regards,

Shaden Kamhawi

co-Editor-in-Chief

Paul Brindley

co-Editor-in-Chief
